Methods

# A targeted sequencing extension for transcript genotyping in single-cell transcriptomics

Lies Van Horebeek[1],*  , Margaux David[1],*  , Nina Dedoncker[1]  , Klara Mallants[1], Baukje Bijnens[1], An Goris[1]  , Bénédicte Dubois[1,2]

As no existing methods within the single-cell RNA sequencing repertoire combine genotyping of specific genomic loci with high throughput, we evaluated a straightforward, targeted sequencing approach as an extension to high-throughput droplet-based single-cell RNA sequencing. Overlaying standard gene expression data with transcript level genotype information provides a strategy to study the impact of genetic variants. Here, we describe this targeted sequencing extension, explain how to process the data and evaluate how technical parameters such as amount of input cDNA, number of amplification rounds, and sequencing depth influence the number of transcripts detected. Finally, we demonstrate how targeted sequencing can be used in two contexts: (1) simultaneous investigation of the presence of a somatic variant and its potential impact on the transcriptome of affected cells and (2) evaluation of allele-specific expression of a germline variant in ad hoc cell subsets. Through these and other comparable applications, our targeted sequencing extension has the potential to improve our understanding of functional effects caused by genetic variation.

## Introduction

Understanding the effect of genetic variants on cellular phenotype can provide new insights into cellular regulatory mechanisms and disease pathogenesis. This is independent of whether these variants are inherited (germline) or acquired (somatic). Genetic variants can modify the cellular phenotype by altering protein sequences, producing alternative transcripts or influencing expression levels. The ability of RNA sequencing (RNA-seq) to capture both genotype and phenotype information renders it an elegant tool for investigating the effects of genetic variation on the transcriptome. A current example is the identification of expression quantitative trait loci across individuals using RNA-seq (Battle et al, 2014; Garrido-Martín et al, 2021). Nonetheless, to avoid the influence of confounders, such

as differences in genetic background and/or environmental factors, on the observed phenotypic changes, large datasets are required for this type of studies. Alternatively, allele-specific expression (ASE)—that is, systematic deviations from the expected 50/50 ratio between parental alleles—through RNA-seq circumvents these confounders by capturing the effect of transcribed variants within heterozygous individuals instead of across persons. Therefore, this approach enables the investigation of *cis*-regulatory mechanisms with a reduced sample size (Li et al, 2012). In addition, as ASE effects are expected to be cell type- and cell state-dependent, moving to single-cell RNA-seq (scRNA-seq) is likely to improve the resolution of these findings (Sandberg, 2014; Song et al, 2017).

Opposed to germline variants, not all cells necessarily carry a specific acquired (somatic) variant, making identification of the affected cells the first obstacle in the investigation of a variant's effect on transcription. However, in the case of transcribed somatic variants, scRNA-seq has emerged as a very useful tool as it enables to simultaneously identify cells harboring the variant and characterize its influence on the transcriptome.

In recent years, a variety of scRNA-seq strategies have become available. Each technique displays unique strengths and limitations with regard to sequence length, number of cells sequenced, and the ability to correct for amplification bias. Full-length scRNA-seq methods such as SMART-seq provide sequence information across the entire length of the transcripts, which allows genotyping and the study of splice isoforms (Picelli et al, 2014). However, the scale of this method is limited to a few hundred cells. In SMART-seq3, the addition of unique molecular identifiers (UMIs) at the 5' ends of transcripts can address amplification bias for quantitative application but only partially, as digital counting does not extend to reads further from the 5' end (Hagemann-Jensen et al, 2020). Alternative, non-full-length methods, such as Drop-seq or 10x Genomics' Chromium scRNA-seq, only provide random sequence information at the 3' or 5' transcript end after isolating single cells through a microfluidics system (Macosko et al, 2015; Zheng et al, 2017). The high throughput of up to hundreds of thousands of cells make these methods suitable for studying rare cell types and the

---

[1]Laboratory for Neuroimmunology, Department of Neurosciences, Leuven Brain Institute, KU Leuven, Leuven, Belgium   [2]Department of Neurology, University Hospitals Leuven, Leuven, Belgium

Correspondence: benedicte.dubois@uzleuven.be
*Lies Van Horebeek and Margaux David share first authorship

use of UMIs enables correction for amplification biases, thereby making quantitative assessments possible.

To supplement the available high-throughput scRNA-seq methodologies with genotyping specific genomic loci, we implemented a straightforward extension. Overlaying unbiased gene expression data (further referred to as the "standard method") with genotype information at the transcript level (from our "targeted method") provides a helpful tool for evaluating the phenotypic impact of genetic variants. Here, we describe the targeted sequencing approach and demonstrate how it can be used to study the effect of somatic variation or to measure ASE.

## Results

### Targeting transcripts and regions of interest in remnants of amplified cDNA

Microfluidics-based scRNA-seq offers high throughput, but its design of sequencing random, short fragments near the 3′ or 5′ end of transcripts is not ideal for applications where efficient genotyping of specific loci within the transcripts is desired. As addition to the 10x Chromium Single Cell Gene Expression and Immune Profiling Solutions, we introduced a targeted sequencing approach in which the user determines which part of the transcript is sequenced with short-read sequencing. Without interfering with the standard method, the targeted sequencing adds an extra layer of information on top of the unbiased gene expression landscape.

The 10x Chromium protocol results in 40 μl amplified cDNA of which up to 20 μl is used for standard library preparation, leaving

remnants available for other applications. From this remaining amplified cDNA, the targeted library is prepared through two rounds of tailed-PCR with only one transcript-specific primer in the first amplification step (Figs 1 and S1). The final construct in the targeted library has the same structure as in the standard library, allowing pooled sequencing of both libraries and data analysis with Cell Ranger. Various primers can be multiplexed in the first amplification step, thereby enabling simultaneous targeting of different regions. As in the standard method, sample indices allow pooling of different samples for sequencing. Primers can be easily adapted to different constructs in newer versions of the standard methodology, or even to other scRNA-seq technologies.

### The standard and targeted outputs may require additional correction of barcodes and UMIs

We applied targeted sequencing in two different contexts: (1) to study a potential effect of a somatic variant on cellular phenotype and (2) to measure ASE at the level of individual cells. For both, we designed one targeted primer to sequence the transcribed variant of interest.

We previously identified a somatic C>G variant at chr3:48508943 (hg19) with a variant allele fraction (VAF, percentage of alleles with alternate genotype) of 2.53% in T cells of a multiple sclerosis patient, corresponding to ~5% of T cells affected by the somatic variant (Van Horebeek et al, 2019). This transcribed, nonsynonymous variant is located 102 base pairs from the 3′ transcript end of TREX1 (NM_033629.6: c.1054C>G p.352L>V). Unpublished data indicate that this variant is enriched in, and possibly even restricted to, the CD8[+] T cell subset. Therefore, we applied standard and targeted

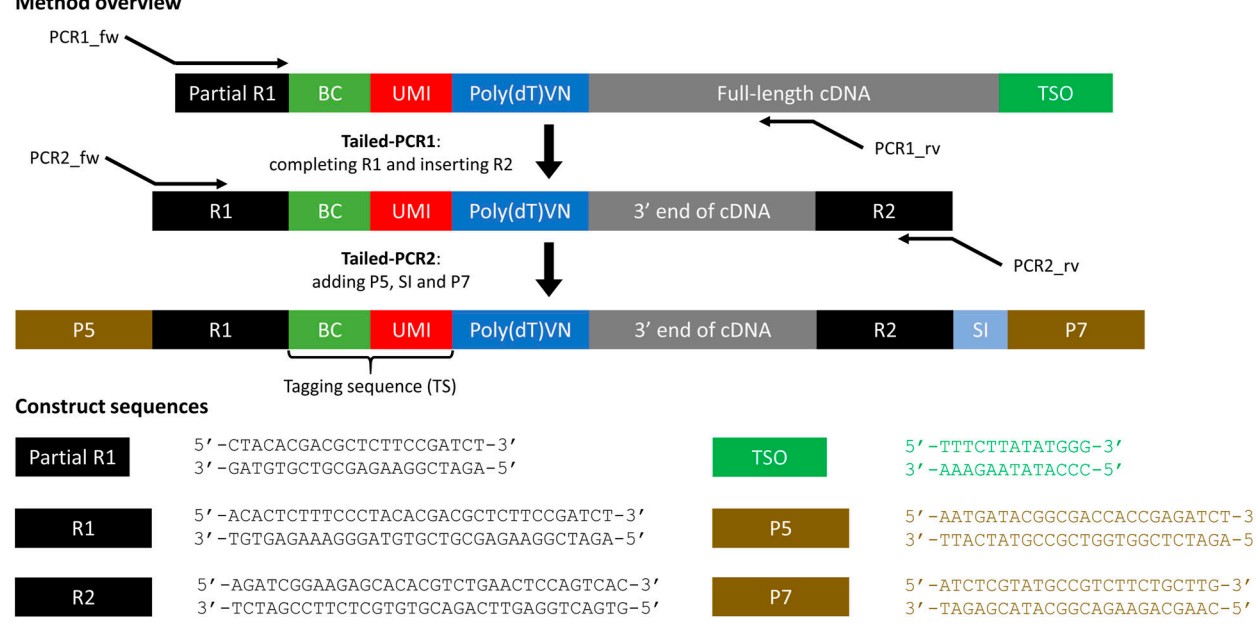

**Figure 1. Targeted library preparation consists of two rounds of tailed-PCR starting from amplified cDNA.**
The constructs and sequences displayed are for extension to the Chromium Single Cell Gene Expression Solution (v3 and v3.1, single index), which captures the 3′ end of transcripts. Image is conceptual and elements are not necessarily proportional. BC, 10x cellular barcode; cDNA, complementary DNA; P5/P7, priming sites used in Illumina sequencers; R1, TruSeq Read 1; R2, TruSeq Read 2; SI, sample index; TSO, template switching oligo; UMI, unique molecular identifier.

scRNA-seq to CD8[+] T cells collected three (T1) and four (T2) years after the sample in which the variant was originally identified (Table S1).

As a proof-of-principle for evaluating ASE, we selected polymorphism rs6897932, associated with multiple sclerosis susceptibility (Gregory et al, 2007) and located in exon 6 of *IL7R*, 817 base pairs from the 5′ transcript end (NM_002185.5: c.818C/T). For this variant, allelic imbalance has previously been detected in bulk RNA-seq data from CD4[+] and CD8[+] T cells (Ban et al, 2020). As switching to the single-cell level may provide greater insights into differences across cell types, we applied the standard and targeted scRNA-seq on T cells of two multiple sclerosis patients (P1 and P2) who are heterozygous for rs6897932.

The sequencing data of the targeted method were processed with the Cell Ranger *count* pipeline. To evaluate whether adapted data processing is necessary, we first plotted the number of reads per tagging sequence (defined as the combination of cellular barcode and UMI) for both the standard and targeted methods (Figs 2A and S2–S5 panels A). Although we saw an overall exponential decrease in tagging sequences with increasing number of reads (e.g., targeted IL7R dataset from P2 shown in Fig 2A), some datasets had a substantial number of tagging sequences supported by only one read (e.g., targeted TREX1 from T1 dataset shown in Fig 2A). The high proportion supported by only one read could be caused by preferential amplification of certain transcripts within the library and/or by an accumulation of amplification and sequencing errors in their tagging sequences, which are not corrected by the Cell Ranger *count* pipeline. To determine the contribution of both possible causes, we evaluated the similarity between tagging sequences of the transcripts of interest via the Hamming distance, that is, the number of dissimilarities between two strings of equal length. For each tagging sequence, we determined the minimal Hamming distance when comparing it with all other tagging sequences found in the dataset (Figs 2B and S2–S5 panels B). Frequently observed tagging sequences are most likely originating from different transcripts, allowing us to determine a reasonable threshold for highly likely independent transcripts. The tagging sequences of these frequently detected transcripts (defined here as covered by >4 reads for standard and >30 reads for targeted method based on our observations) have Hamming distances of five or more between them (Figs 2C and S2–S5 panels C). Thus, tagging sequences with a Hamming distance below 5, especially with Hamming distances of 1 or 2, most likely originate from the same transcript. However, even though highly unlikely, we cannot exclude the possibility that these could capture truly different transcripts with, by chance, very similar tagging sequences.

In the targeted sequencing datasets, >80% of tagging sequences supported by one read have a minimal Hamming distance (compared to all other corrected tagging sequences in the dataset) of five or higher, suggesting these are independent transcripts captured by only one read (Figs 2C and D and S2–S5 panels C and D). Of note, even for T1, the standard method shows indications of artefact accumulation, underlining that this is not specific to the targeted method, but rather depends on the amount of (unequal) amplification taking place. The Cell Ranger *count* pipeline already removes a substantial proportion of artefacts, as before correction by Cell Ranger (Figs S6–S9) as low as

20% of tagging sequences supported by one read have a minimal Hamming distance of five or higher. Depending on the application and the extent of observed artificial increase, additional correction of the tagging sequences are based on the Hamming distances, as detailed in the guidelines based on our data (Supplemental Data 1).

## Technical parameters determine number and proportion of independent transcripts

We expected the number of transcripts and their artificial increase to depend on the amount of amplified cDNA used as input, the number of PCR cycles, and the sequencing depth. To measure their individual impact, we set up a systematic analysis starting from B cells from an untreated multiple sclerosis patient and focusing on the extension to the 10x Genomics 3′ capture method. Five amplified cDNA samples were used to create three libraries per sample, each simultaneously targeting CCL5, CD38, ICOSLG, S100A11, and TREX1 transcripts, and either varying in the number of PCR cycles or in the amount of input cDNA. The targeted transcripts have different expression levels in B cells in multiple sclerosis patients and we were able to successfully target transcripts with gene expression levels as low as ~7 UMIs per 1 million UMIs. Despite their expression levels being similar to the other targets, the number of captured CD38 and ICOSLG transcripts was too low for meaningful analysis and we excluded them from the systematic analysis. For these two transcripts, we were not able to identify the transcripts' sequences in the output from the Sanger sequencing during primer testing. The observed consistency between the optimization results and the systematic analysis validates the use of the (cheaper and faster) Sanger sequencing method during optimization.

We down-sampled the sequencing data to (i) equal sequencing depth datasets for evaluation of the effect of the number of PCR cycles and the amount of input DNA and (ii) 1:100, 1:10, 1:1 of total sequencing data to evaluate the effect of sequencing depth (Table S2). Building upon our previous observations, we corrected tagging sequences with an unacceptably high probability of capturing the same transcript (hamming distance ≤3) to the barcode and UMI supported by most sequencing reads and assumed that this led to a set of tagging sequences each capturing an independent transcript (further referred to as "independent tagging sequences"). To evaluate the influence of the different technical parameters, we compared the absolute number and proportion of independent tagging sequences, the mean number of independent tagging sequences per cellular barcode, and the number of barcodes. As expected, increasing amounts of input cDNA were associated with a higher number and a higher proportion of independent tagging sequences (Fig 3A). Furthermore, increasing the number of PCR cycles substantially increased the number of independent tagging sequences, whereas no systematic changes in the number of dependent tagging sequences were observed, resulting in an unexpected increase in the proportion of independent tagging sequences (Fig 3B). Increasing sequencing depth resulted in higher absolute numbers, but lower proportions of independent tagging sequences (Fig 3C). Across comparisons, increases in the number of independent tagging sequences were associated with an increase

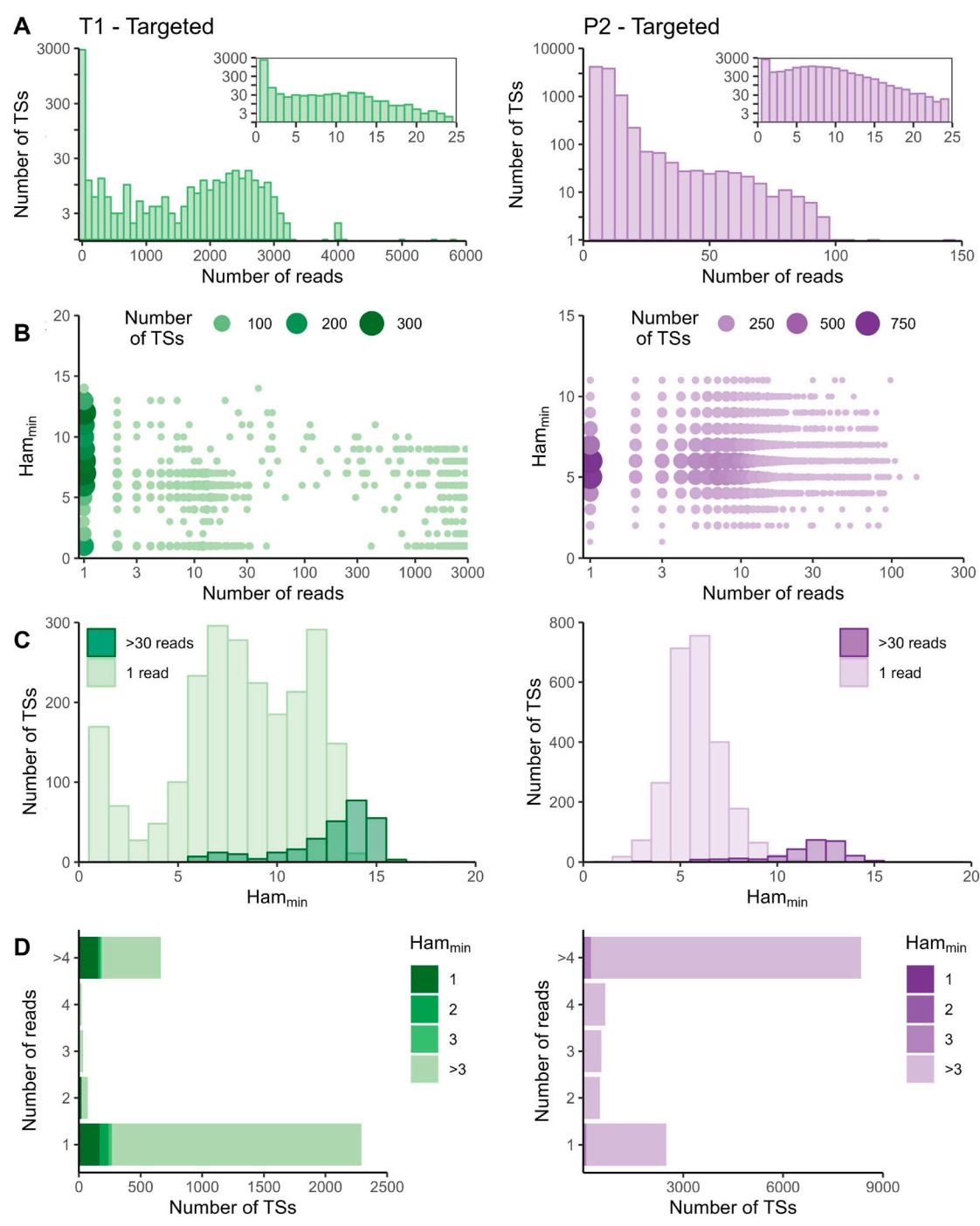

**Figure 2. Examples of read and Hamming distance distributions.**
Plots on the left show the TREX1 data from the targeted method at T1. Plots on the right the IL7R data from the targeted method for P2. **(A)** Distribution of number of reads per tagging sequence from the targeted method (left: bin width = 100 for the base plot and bin width = 1 for the zoomed plot; right: bin width = 5 for the base plot and bin width = 1 for the zoomed plot). **(B)** Count plot of tagging sequences by the number of reads and minimal Hamming distance. **(C)** Histograms of minimal Hamming distance for tagging sequences supported by only one read (light; compared with the whole dataset) and amongst abundantly present tagging sequences (dark). **(D)** Bar chart of corrected tagging sequences per number of reads and colored by minimal Hamming distance (to all other tagging sequences). $Ham_{min}$, minimal Hamming distance; TS, tagging sequence.

in barcodes and a variable effect on the mean number of transcripts per barcode (Fig 3A–C).

These datasets also allowed us to evaluate the amount of extra information obtained from (i) sequencing new targeted libraries created from the same amplified cDNA and (ii) additional sequencing of existing targeted libraries. For the first aim, we determined the overlap in captured CCL5, S100A11, and TREX1 transcripts between the libraries with varying amounts of input

**Figure 3. Systematic analysis reveals the influence of the technical parameters and shows the potential value of additional libraries and sequencing.**
**(A, B, C)** The effect of amount of input cDNA (A), the number of PCR cycles (B), and sequencing depth (C) on the number of independent tagging sequences (iTSs), proportion of iTSs compared with all observed TSs, mean number of iTSs per barcode (BC), and number of BCs. Shape indicates transcript (circle: CCL5, triangle: S100A11, square: TREX1), color indicates cDNA sample, and line type (only in (C)) indicates sequencing library. **(D)** Overlap in transcripts between libraries generated from the same

cDNA from the same sample but with the same number of PCR cycles and reads (Figs 3D and S10). In each dataset, 68.9–90.3% of transcripts are unique for that dataset, with systematically higher percentages and higher absolute numbers obtained in datasets with more input cDNA. For the second aim, one dataset was randomly down-sampled to 21 datasets, that is, three replicates each of seven different sequencing depths. At each depth, the overlap in CCL5, S100A11, and TREX1 transcripts between the three replicates was determined (Figs 3E and S11) and we calculated that across transcripts and sequencing depths, 21.0–53.4% of transcripts captured in a dataset were unique for that replicate. The sequencing saturation as estimated by Cell Ranger ranged from 75.2–75.3% for the smallest datasets (600,000 reads) to 87.6% for the largest datasets (4,200,000 reads).

### Targeted sequencing increases detection of TREX1 transcripts, but not of IL7R transcripts

Next, to quantitatively compare standard and targeted sequencing, we returned to the TREX1 and IL7R datasets and restricted our analyses to data from live, single cells. Based on our previously described observations (Figs S2–S5), we implemented the additional correction based on Hamming distance for the TREX1 dataset (threshold for correction is a Hamming distance ≤3), but not for the IL7R dataset. Without this correction, the number of TREX1 transcripts and cells would have been artificially increased by up to 11% (Table 1).

Although the number of detected TREX1 transcripts per cell was correlated between standard and targeted methods (Fig S12), the targeted method detected 8x (T1) and 12x (T2) as many TREX1 transcripts in 4x (T1) and 6x (T2) as many cells as the standard method. More than 85% of transcripts detected with the standard method are also detected with the targeted approach. The targeted method detected slightly lower numbers of IL7R transcripts and cells with transcripts than the standard approach. Nevertheless, the number of detected IL7R transcripts per cell showed an even stronger correlation between standard and targeted methods than for TREX1, with ~50% of captured IL7R transcripts shared between both methods (Fig S13).

### Genotyping transcribed variants in individual transcripts

Whereas standard scRNA-seq sequences a random fragment close to either the 3′ or 5′ end, targeted sequencing can be designed to sequence over a specific region of interest, for example, to systematically determine genotypes for transcribed variants in individual transcripts. With the targeted method, even a low number of reads per transcript (median of 1 read per transcript) resulted in genotype information for ~70% of TREX1 transcripts and 98.9% of cells with a TREX1 transcript (Table 1). Combined with the increased capture of TREX1 transcripts, the number of transcripts with

genotype information in the targeted method was one order of magnitude larger compared with the standard method (9.5x for T1, 27.8x for T2).

Even though the targeted method did not capture more IL7R transcripts, the more efficient genotyping (75% of transcripts, >95% of cells with transcripts) resulted in an increase with two orders of magnitude in the number of transcripts with genotype information compared with the standard method (104x for P1, 102x for P2, Table 1). The median of seven reads per transcripts and five (P2) or six (P1) transcripts per cell also offers the possibility to increase confidence in genotype calls by demanding a minimum number of reads per transcript and to estimate ASE more accurately by implementing a threshold in the number of transcripts per cell, both of which have been previously suggested (Prashant et al, 2021).

Amplification and sequencing of transcripts are not error-free, but the tagging sequences allow estimation of the error rate. Including transcripts with two or more reads containing genotype information, we estimate error rates at $2.81 \times 10^{-4}$ (TREX1–T1), $6.83 \times 10^{-4}$ (TREX1 – T2), $1.98 \times 10^{-2}$ (IL7R–P1), and $1.05 \times 10^{-2}$ (IL7R–P2) and the proportion of reads without genotype information at $1.35 \times 10^{-3}$ (TREX1–T1), $2.39 \times 10^{-3}$ (TREX1–T2), $5.12 \times 10^{-3}$ (IL7R–P1), and $5.56 \times 10^{-3}$ (IL7R–P2).

Preferential amplification of one of the alleles can occur when alternate genotypes are associated with transcripts with a different length or with a nearby variant located in the primer-binding region, impeding quantitative analyses based on the genotype. This preferential amplification, resulting in amplification bias, can be assessed by comparing the distribution of number of reads per tagging sequence for all genotypes. For TREX1 and IL7R, we observed no significant differences between the number of reads covering a specific allele and thereby conclude that there was no preferential amplification related to the genotype (Table 1).

### Targeted sequencing for detection of cells affected by somatic variants

Combining information from the standard and targeted methods, genotype information was available for 524 and 2,587 cells from T1 and T2, respectively (Fig 4B and C). The alternate allele was detected in 1.72% and 1.16% of genotyped cells from T1 and T2. The corresponding VAFs, of 0.86% and 0.58% in the CD8$^+$ T cells were remarkably lower than the 2.53% VAF in the original T cell sample (including CD4$^+$ and CD8$^+$ subset), suggesting that this clone may have decreased in size over time. The cells in which we detected the alternate allele are similarly distributed across different cell subsets (Fisher's exact test for count data: $P = 0.11$ for distribution of cells with alternate alleles versus genotyped cells across clusters) (Fig 4A–C). No substantial gene expression changes within clusters or systematic changes across clusters could be detected when comparing cells in which the alternate allele was detected to cells in which it was not.

cDNA sample. iTSs unique to a library are indicated in dark orange, and iTSs shared with at least one other library obtained from the same sample in light orange. **(E)** Overlap in transcripts between replicates at different sequencing depths (ranging from 600,000 reads to 4,200,000 reads). iTSs unique to a replicate are indicated in dark orange, and iTSs shared with at least one other replicate with the same number of reads in light orange.

**Table 1.  Number of (genotyped) transcripts of interest and cells with (genotyped) transcripts.**

| | T1 | T1 | T2 | T2 | P1 | P1 | P2 | P2 |
|---|---|---|---|---|---|---|---|---|
| | standard | targeted | standard | targeted | standard | targeted | standard | targeted |
| **Reads** | | | | | | | | |
| Target | TREX1 | TREX1 | TREX1 | TREX1 | IL7R | IL7R | IL7R | IL7R |
| N mapped reads | 188,016,191 | 961,659 | 207,371,231 | 1,642,892 | 135,115,396 | 200,707 | 124,395,202 | 104,123 |
| N on-target reads | 4,500 | 795,850 | 3,994 | 1,373,821 | 129,577 | 191,280 | 83,925 | 98,789 |
| Percentage on-target (%) | $2.39 \times 10^{-3}$ | 82.76 | $1.92 \times 10^{-3}$ | 83.62 | $9.59 \times 10^{-2}$ | 95.30 | $6.57 \times 10^{-2}$ | 94.88 |
| **Transcripts** | | | | | | | | |
| All | 167 | 1,452 | 483 | 6,106 | 23,906 | 18,048 | 13,384 | 9,853 |
| Independent[a] | 163 | 1,316 | 483 | 5,964 | NA | NA | NA | NA |
| Increase (%) | 2.45 | 10.33 | 0.00 | 2.38 | NA | NA | NA | NA |
| N reads: median; mean | 11; 14.60 | 1; 193.00 | 2; 3.10 | 1; 40.90 | 4; 4.08 | 7; 8.12 | 4; 4.58 | 7;7.78 |
| Genotyped | 95 | 908 | 150 | 4,187 | 137 | 13,926 | 71 | 7,368 |
| Genotyped (%) | 58.28 | 69.00 | 31.06 | 70.20 | 0.57 | 77.16 | 0.53 | 74.78 |
| N ref | 93 | 898 | 149 | 4,158 | 68 | 6,284 | 30 | 3229 |
| N reads ref: median (range) | 17 (1–65) | 1 (1–5,809) | 3 (1–15) | 1 (1–1,439) | 4 (1–10) | 8 (1–122) | 5 (1–12) | 8 (1–105) |
| N alt | 2 | 10 | 1 | 29 | 69 | 7,642 | 41 | 4,139 |
| N reads alt: median (range) | 20.5 (16–25) | 1 (1–2,579) | 4 (4–4) | 1 (1–802) | 4 (1–16) | 8 (1–223) | 5 (2–11) | 7 (1–146) |
| **Cells** | | | | | | | | |
| All | 131 | 577 | 446 | 2,684 | 3,609 | 3,342 | 2,551 | 2,278 |
| Independent[a] | 129 | 528 | 446 | 2,613 | NA | NA | NA | NA |
| Increase (%) | 1.55 | 8.49 | 0.00 | 2.72 | NA | NA | NA | NA |
| N transcripts: median (range) | 1 (1–2) | 2 (1–10) | 1 (1–2) | 2 (1–10) | 1 (1–2) | 6 (1–23) | 1 (1–2) | 5 (1–23) |
| Genotyped | 91 | 523 | 146 | 2,584 | 136 | 3,259 | 68 | 2,183 |
| Genotyped[b] (%) | 70.54 | 99.05 | 32.74 | 98.89 | 3.77 | 97.52 | 2.67 | 95.83 |

N reads, number of reads per transcript; N ref/alt, number of transcripts with respectively reference or alternate allele called; N reads ref/alt, number of reads per transcript for transcripts with respectively reference or alternate allele called. N transcripts: number of transcripts per cell.
[a]Reference for percentages Increase and Genotyped in case of extra correction based on Hamming distance (TREX1 data only).
[b]At least one transcript has genotype information available. Only transcripts from live, single cells are included.

### Evaluating ASE

Combining genotype data from the standard and targeted sequencing for the rs6897932 variant in *IL7R* resulted in 3,259 and 2,183 high-quality cells from P1 and P2, respectively. The risk allele fraction (RAF, proportion of transcripts carrying the risk allele) could not be accurately determined using only data from the standard method, as most of the cells had genotype data for only one transcript. This is reflected in RAF estimates (0 and 1) (Fig 5B). The inaccuracy resulting from the lack of data masked allelic imbalance for P1 (average RAF including all transcripts and cells: 0.50 [P1, based on 136 cells] and 0.43 [P2, based on 68 cells]).

The targeted method increased both the number of genotyped transcripts per cell and the number of cells with genotype information, enabling us to more accurately estimate RAF for individual cells and for clusters of cells (Fig 5A and C). When we applied a previously suggested threshold of minimum five genotyped transcripts per cell (minT = 5, Prashant et al, 2021) we obtained an average RAF of 0.45 for CD4[+] T cells (based on 1,255 cells across

both patients, $P = 4.41 \times 10^{-4}$ for deviation from 50/50 ratio) and 0.46 for CD8[+] T cells (based on 503 cells, $P = 0.08$). These estimates are identical to findings from bulk RNA-seq (Ban et al, 2020). Depending on the application, a less strict minT threshold can be considered, if the less precise RAFs for individual cells are compensated by including more cells (Table S3). With minT = 3, we estimate average RAFs of 0.44 for CD4[+] T cells (based on 2,158 cells, $P = 2.81 \times 10^{-8}$) and 0.45 for CD8[+] T cells (based on 957 cells, $P = 2.20 \times 10^{-3}$).

## Discussion

We present a straightforward and flexible targeted sequencing method that substantially increases the amount of information obtained from classical single-cell transcriptomics. We are aware of one similar approach that was published during our development phase (Nam et al, 2019), but this approach shows two important methodological differences with ours. First, Nam et al (2019) added

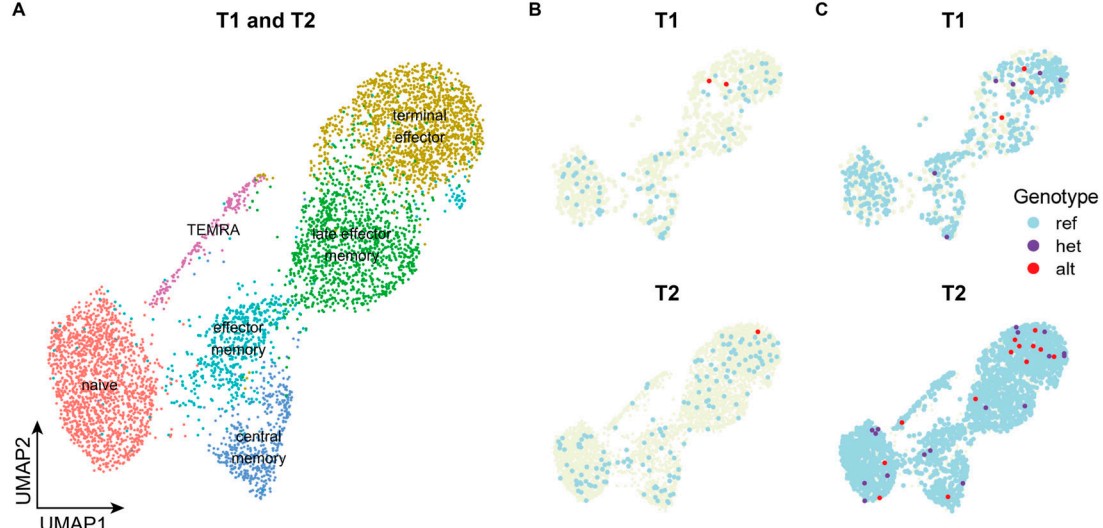

**Figure 4. Targeted sequencing substantially increases the number of transcripts and cells with genotype information.**
**(A)** Clustering and annotation based on gene expression data from the standard method of live, single cells. **(B, C)** Genotype information obtained through the standard method (B) and the targeted method (C). Alt: cells in which only alternate transcripts were detected, het: cells in which reference transcripts and alternate transcripts were detected, ref: cells in which only reference transcripts were detected, TEMRA: terminally differentiated effector memory rexpressing CD45RA.

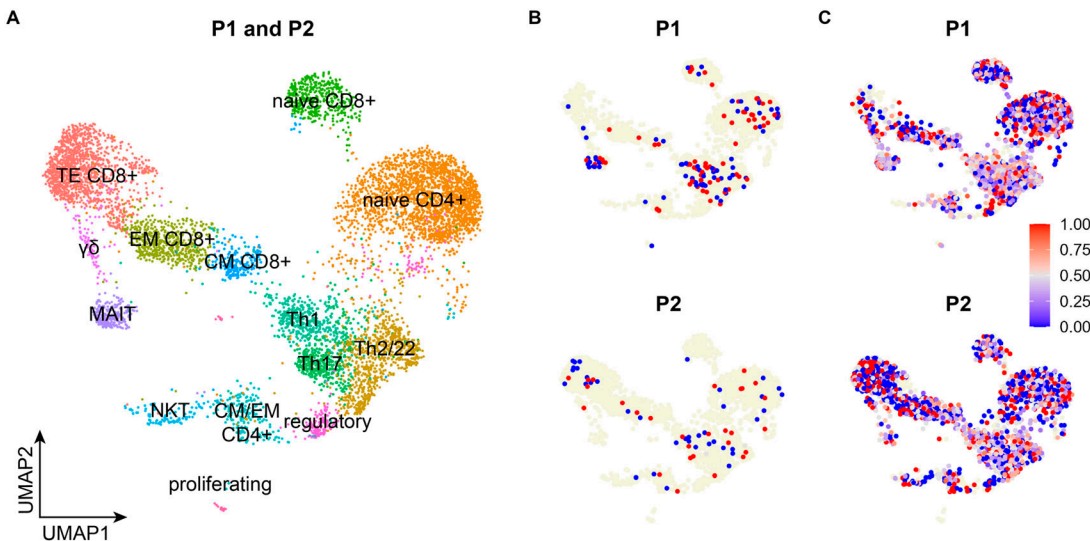

**Figure 5. Targeted sequencing substantially increases the number of cells with genotype information and improves risk allele fraction estimates.**
**(A)** Clustering and annotation based on gene expression data from the standard method of live, single cells from a larger cohort of MS patients (n = 9) and down-sampled to the relevant subsets (P1 and P2). **(B, C)** Risk allele fraction estimates obtained from the standard method (B) and the targeted method (C). CM, central memory; EM, effector memory; MAIT, mucosa-associated invariant T cells; NKT, natural killer T cells; RAF, risk allele fraction; TE, terminal effector; Th, T helper cells.

transcript-specific spike-ins during reverse transcription. This may increase the number of transcripts captured and detected but can also introduce biases. Working without spike-ins offers the advantage of being able to add targeted sequencing to previous experiments for which cDNA was stored and to flexibly decide which transcripts are of interest after analyzing data from the standard method. Second, the primers used in our approach are simpler in structure—without staggers (i.e., nucleotide sequences of variable length)—leading to easier design and resulting in more flexibility for

adaptations to other 10x Chromium versions or even other scRNA-seq methodologies. Pooling and/or increasing library complexity artificially (e.g., higher amount of PhiX) can compensate for lower diversity in the targeted library in our approach.

Optimization steps are required during primer design and testing, and are recommended to determine a good balance between amount of input cDNA, number of PCR cycles, and sequencing depth. As demonstrated by our systematic analysis, these three parameters affect the number of detected transcripts and the

accumulation of errors in barcodes and UMIs, which may lead to a perceived increase in transcripts if adequate data processing is lacking. More input cDNA increases the number of detected transcripts while reducing artificial increase, but is often capped by the limited amount available and its use in other applications such as T- or B-cell receptor sequencing. Increasing the number of PCR cycles within the tested range leads to more transcripts acquiring the construct structure required for sequencing and thus more transcripts being detected and less artificial increase. Further increasing the number of PCR cycles may saturate new transcript detection (finite number of transcripts) while increasing the accumulation of amplification artefacts, tipping the balance towards amplifying artificial increase. An alternative approach to limit unequal amplification between transcripts is switching to linear amplification, as demonstrated in Constellation-Seq for 20 target genes simultaneously, but without genotyping of transcribed variants (Vallejo et al, 2021). Furthermore, increasing sequencing depth unsurprisingly results in more detected transcripts. However, increased sequencing depth also results in easier detection of amplification artefacts and more sequencing artefacts, magnifying artificial increase. In addition, the data from the systematic analysis indicate that, if necessary, the number of detected transcripts can be substantially increased afterward, by creating a new library from the same cDNA sample or by additional sequencing of an existing library, even when the sequencing saturation calculated by Cell Ranger reaches up to 90%.

Targeted sequencing may substantially increase the number of detected transcripts up to an order of magnitude, as observed in the TREX1 data. No increase in IL7R transcripts was observed, most likely because of the location of the transcript-specific primer in exon 6. This exon is skipped in approximately half of IL7R transcripts (median [IQR]: 45% [40–50%] in 30 controls, 50% [45–55%] in 23 multiple sclerosis patients) (Cardamone et al, 2019). Transcripts where exon 6 is spliced out cannot be detected by the targeted approach, offering a biological explanation for the targeted approach missing 42% of IL7R transcripts detected by the standard method. However, we cannot rule out that technical issues are contributing to the lower number of IL7R transcripts. The higher median number of sequencing reads covering each transcript in the targeted datasets indicates sequencing depth was sufficient, but we cannot exclude the possibility that primer concentrations were not sufficiently high for the very abundant IL7R transcripts.

When investigating a transcribed variant, targeted sequencing can increase the number of genotyped transcripts with up to two orders of magnitude compared with the standard method. Transcript tagging with barcodes and UMIs allows estimation of the error rate for the transcribed variants. In our datasets, the error rate was sufficiently low for the TREX1 variant, with fewer than 0.07% of reads having a wrong genotype call. The error rate was substantially higher for the IL7R variant at up to 2% of reads with a wrong genotype call, but this is not expected to be problematic as most transcripts have genotype information from multiple reads. In case the error rate is unacceptably high for the chosen application, a minimal threshold of reads supporting a genotype can be implemented, at the cost of fewer genotyped transcripts.

Adding a targeted sequencing layer to a high-throughput single-cell transcriptomics method opens up possibilities for several applications, of which we demonstrated two. First, we showed how to apply targeted sequencing for identification of cells with transcribed somatic variants. Unbiased gene expression data of the same cells enable linking somatic variants to gene expression changes, aiding to discern whether and if so, which effect the variant has on cellular phenotype. Here, we genotyped >3,000 CD8⁺ T cells of a multiple sclerosis patient for the TREX1 c.1054C>G variant, but could not link genotype to transcriptional alterations, suggesting the variant does not have a major impact on cellular phenotype under these conditions. Second, we demonstrated with variant rs6897932 in IL7R how to evaluate ASE for transcribed variants in heterozygous individuals ex vivo. We calculated the fraction of risk allele per cell and used cell-type annotations from the standard method to evaluate allelic imbalance in cell types of interest. Our estimates for allelic imbalance in CD4⁺ and CD8⁺ T cells correspond to previous estimates based on bulk data (Ban et al, 2020). Although any characteristic may be used to identify groups of interest, sufficient cells in each group are needed to level out stochastic effects, for example, transcriptional bursting. Application of this method to transcribed variants in high linkage disequilibrium with GWAS risk signals may facilitate fine-mapping and increase our understanding of how genetic variants alter disease susceptibility.

Unfortunately, this targeted sequencing method also comes with a number of limitations. Primer design and optimization steps are labor intensive and sometimes difficult to evaluate. We attempt to facilitate these steps by sharing our protocols to prepare test cDNA and for primer design and testing (Supplemental Datas 2 and 3). Transcript expression levels and capture efficiency place a boundary on the number of transcripts that can be assessed by targeted sequencing. Biochemical improvements in updated 10x Chromium chemistries are expected to lead to higher portions of captured transcripts. Targeted sequencing in the format we describe here is expected to be most efficient for regions and transcribed variants close to the transcript ends (here, up to ~1,000 nucleotides), as longer fragments are captured less efficiently on Illumina sequencers. One solution for variants positioned further from the 3′ or 5′ end is to remove the superfluous middle part of the construct, as done in the circular method from Nam et al (2019).

In summary, we described a new targeted sequencing method, formulated recommendations for its implementation, and provide two proof-of-concepts for applications. This method offers a helpful step forward in evaluating the effect of transcribed genetic variants—whether germline or somatic—on the transcriptome of single cells in an unbiased way.

# Materials and Methods

### Patient inclusion and sample selection

Multiple sclerosis patients diagnosed based on the 2017 revised McDonald criteria (Thompson et al, 2018) were recruited from the University Hospitals Leuven (UZ Leuven), Belgium. The study has been approved by the Ethics Committee of the University Hospitals Leuven (ML4733), and written informed consent was obtained from

all participants. In total, our cohort consisted of four multiple sclerosis patients: three untreated patients (P1 and P2 who were heterozygous for rs6897932 in *IL7R* and one for the systematic analysis), and a fourth patient from a previous study, in which we identified a somatic variant in *TREX1* (NM_033629.6: c.1054C>G p.352L>V) close to the 3′ transcript end (Van Horebeek et al, 2019). Peripheral blood samples were collected in 10-ml blood tubes containing EDTA (BD Vacutainer). PBMCs were isolated using Lymphoprep (Axis-Shield), resuspended in 1 ml FBS (Tico Europe) containing 10% DMSO (Sigma-Aldrich) and stored in liquid nitrogen until use.

### Single-cell transcriptomics—standard method

PBMCs were thawed on ice and washed with 2% FBS in PBS containing 2 mM EDTA. B cells, T cells or CD8⁺ T cells were enriched by immunomagnetic negative selection using EasySep Human B Cell, T or CD8⁺ T Cell Isolation Kits (STEMCELL Technologies), respectively, according to the manufacturer's instructions. Single-cell suspensions were used for single-cell transcriptomics with the Chromium Next GEM Single Cell 3′ Reagent Kits v3.1 or the Chromium Single Cell V(D)J Reagent Kits v.1.1 (10x Genomics) following the manufacturer's protocol. Libraries were sequenced with Illumina NovaSeq technology (Nucleomics Core, KU Leuven).

### Targeted sequencing

#### Test cDNA for primer optimization

As only 40 $\mu$l amplified cDNA is obtained from the standard method, we prepared large quantities of test cDNA with the same construct structure as amplified cDNA (first constructs in Figs 1 and S1) to be used for primer optimization. Test cDNA was prepared from RNA extracted from the cell subset of interest through reverse transcription (RT) and amplification. Different primers and template switching oligos were used for 3′ and for 5′ test cDNA (Table S4). The reverse transcription mixture contained 1x Maxima RT buffer, 1 mM dNTPs, RNase inhibitor, 20 $\mu$M poly(dT) primer, 5 $\mu$M template switching oligo, Maxima H minus RTase, and 15 ng/$\mu$l RNA. Mixture of RNA and poly(dT) primer was prepared first and added last to the reverse transcription mixture. Reverse transcription took place at 50°C during 30 min, followed by 5 min of enzyme inactivation at 85°C. Remaining primers were digested using Exosap-IT (Thermo Fisher Scientific). The resulting cDNA was amplified using PCR. The reaction mix consisted of 1x KAPA HiFi Hotstart ReadyMix (Roche), 0.3 $\mu$M "forward primer," 0.3 $\mu$M "reverse primer" and 1:2 cleaned-up cDNA. Initial denaturation took place during 3 min at 95°C, followed by 25 cycles of denaturation during 20 s at 98°C, annealing at 65°C during 15 s, and extension at 72°C during 3 min. Final extension took place at 72°C during 5 min. Amplified cDNA was purified using SPRI (Beckman) clean-up (1.0x) and stored immediately at 4°C for short term usage or −20°C for medium-term usage (up to 6 mo).

#### Primer optimization

Primers were designed following a predesigned template (Supplemental Data 3), using PrimerBLAST (Ye et al, 2012). Potential primers were tested by using them individually in the first tailed-PCR of the targeted library preparation with either 3′ or 5′ test cDNA,

with 30 cycles and a temperature gradient for the annealing temperature. Primers were initially evaluated by analyzing fragment sizes in the PCR product using Bioanalyzer (Agilent) or TapeStation (Agilent). If a fragment of the expected length was detected, the correct sequence was confirmed using Sanger sequencing (LGC Genomics). Primers for multiplex reactions were first optimized individually and then tested in multiplex in a similar setup.

#### Targeted library preparation

A targeted library is prepared from amplified cDNA by two subsequent tailed PCRs (Figs 1 and S1 and Table S5). The first PCR mix consisted of 1x KAPA HiFi HotStart ReadyMix, 0.3 $\mu$M forward primer PCR1, 0.3 $\mu$M of each (transcript-specific) reverse primer, and amplified cDNA left over from the standard method. As default, we used 10 $\mu$l amplified cDNA in 25 $\mu$l reaction volume. We prepared libraries starting from 5, 10 or 20 $\mu$l of amplified cDNA in 50 $\mu$l reaction volume to evaluate the effect of input amount. As default, initial denaturation took place during 3 min at 95°C, followed by 15 cycles of denaturation during 20 s at 98°C, annealing at 65°C during 15 s, and extension at 72°C during 30 s. We varied the number of cycles between 10, 15, and 20 cycles to evaluate the effect of number of PCR cycles. Final extension took place at 72°C during 1 min. PCR products were purified using SPRI clean-up (1.2x, 20 $\mu$l elution volume). The second PCR mix consisted of 1x KAPA HiFi HotStart ReadyMix, 0.3 $\mu$M forward primer PCR2, 0.3 $\mu$M reverse primer PCR2, and cleaned up the product from PCR1. We used 20 $\mu$l purified PCR1 product in 50 $\mu$l reaction volume. Thermal conditions were identical to the first PCR. PCR products were purified using SPRI clean-up (1.0x, 20 $\mu$l elution volume). Targeted libraries were sequenced with 10% PhiX with the Illumina NextSeq500 technology (Nucleomics Core, KU Leuven) or Illumina MiSeq technology (CeGaT GmbH), with the following read lengths: R1: 28, i7: 8, i5: 0, R2: 150.

### Data analysis

Random down-sampling of sequencing data was done with seqtk-1.3. All datasets (both standard and targeted) were analyzed independently using Cell Ranger 5.0.1 and 6.1.1 (only IL7R data) pipelines. Filtered gene expression matrices from the standard method were further processed in R with the *Seurat* package (version 4.3.0). Gelbeads-in-emulsion with 800–2,500 different genes and at most 10% mitochondrial RNA were considered live, single cells. Quality control of the IL7R datasets was performed with same parameters, except nFeatures which was set to >500 and doublets were removed using *scDblFinder* (v1.12.0). The IL7R datasets (P1 and P2) were integrated with other T cell scRNA-seq (total n = 9) from multiple sclerosis patients to improve the quality of clustering and annotation. The TREX1 datasets (T1 and T2) were also merged for quality control, clustering, and annotation. Clusters were annotated with the automatic annotation tool *SingleR* (v2.0.0), with *celldex*'s (v.1.8.0) MonacoImmuneData as reference, before these annotations were manually refined using the *Azimuth* platform (Hao et al, 2021) and markers from two publications (Szabo et al, 2019; Ostkamp et al, 2022). Raw and corrected barcodes and UMIs of transcripts aligned to regions of interest were extracted from the outputted BAM files. Genotypes were determined at the

level of individual reads and transcripts using VarTrix v1.1.3 (for all other datasets) and v1.1.22 (10X Genomics) for the IL7R data, which was run in debugging mode with UMI correction. Genotype information from individual transcripts was extracted from the debugging file. Further data analyses and visualization were performed in R version 4.4.2, using R packages *dplyr* (v1.1.1), *stringr* (v1.5.0), *tidyr* (v1.3.0), *data.table* (v1.14.8), *table1* (v1.4.3), *ggplot2* (v3.4.2), and *ggVennDiagram* (v1.2.2). Hamming distances were calculated using the *StrDist* function from R package *DescTools* (v0.99.48).

For the TREX1 and systematic analysis datasets, tagging sequences (i.e., barcodes and UMIs) of reads that mapped to the same transcript and with a hamming distance ≤3 were corrected to the barcode and UMI for which the data provide most evidence, that is, highest number of supporting reads. Datasets from the same amplified cDNA were analyzed together and the number of reads supporting a barcode or UMI was normalized to account for differences in sequencing depth between datasets. Sets of tagging sequences requiring correction were identified by calculating Hamming distances to all other tagging sequences, starting with the tagging sequences supported by the highest number of reads (or highest weight if normalization for dataset size is applicable).

## Data Availability

All processed sequencing data generated in this study have been submitted to the NCBI Gene Expression Omnibus under accession number GSE223704. Raw sequencing data are available through the corresponding author upon request. Protocols, guidelines, and code snippets for analysis of the targeted sequencing data are included in the Supplemental Datas 1–4.

## Supplementary Information

## Acknowledgements

L Van Horebeek received a PhD Fellowship from the Belgian Charcot Foundation. M David holds a PhD Fellowship Fundamental Research from the Research Foundation – Flanders (FWO). B Dubois is a Clinical Investigator (BOF-FKO) at KU Leuven. This research was supported by the Research Fund KU Leuven (C24/16/045) and the Research Foundation – Flanders (FWO GOA7219N). Some of the resources and services used in this work were provided by the VSC (Flemish Supercomputer Center), funded by the Research Foundation – Flanders (FWO) and the Flemish Government. The graphical abstract was created with icons from BioRender.com. We thank Suresh Poovathingal for assistance with setting up the single-cell experiments and Jarne Beliën, Dries De Wit, and Stijn Swinnen for critical revision of our article. Finally, the authors thank Katleen Clysters and Cindy Thys for their contribution in sample collection, and all patients and their families for their willingness to participate in this study.

## Author Contributions

L Van Horebeek: conceptualization, resources, data curation, formal analysis, funding acquisition, visualization, methodology, and writing—original draft, review, and editing.
M David: data curation, formal analysis, funding acquisition, visualization, methodology, project administration, and writing—original draft, review, and editing.
N Dedoncker: data curation and methodology.
K Mallants: data curation and methodology.
B Bijnens: data curation and writing—review and editing.
A Goris: conceptualization, supervision, funding acquisition, methodology, writing—original draft, and project administration.
B Dubois: data curation, supervision, funding acquisition, project administration, and writing—review and editing.

## Conflict of Interest Statement

The authors declare that they have no conflict of interest.

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
