## [Reviewer comments · Life Science Alliance]

Life Science Alliance

A targeted sequencing extension for transcript genotyping in single-cell transcriptomics

Lies Van Horebeek, Margaux David, Nina Dedoncker, Klara Mallants, Baukje Bijmens, An Goris, and Benedicte Dubois
DOI: <https://doi.org/10.26508/lsa.202301971>

Corresponding author(s): *Benedicte Dubois, KU Leuven*

Review Timeline:	Submission Date:	2023-02-05
	Editorial Decision:	2023-03-17
	Revision Received:	2023-07-14
	Editorial Decision:	2023-08-07
	Revision Received:	2023-08-17
	Accepted:	2023-08-29

Transaction Report:

March 17, 2023

Re: Life Science Alliance manuscript #LSA-2023-01971-T

Lies Van Horebeek
Katholieke Universiteit Leuven

Dear Dr. Van Horebeek,

Thank you for submitting your manuscript entitled "A targeted sequencing extension for transcript genotyping in single-cell transcriptomics" to Life Science Alliance. The manuscript was assessed by expert reviewers, whose comments are appended to this letter. We invite you to submit a revised manuscript addressing the Reviewer comments.

Thank you for this interesting contribution to Life Science Alliance. We are looking forward to receiving your revised manuscript.

Sincerely,

B. MANUSCRIPT ORGANIZATION AND FORMATTING:

Reviewer #1 (Comments to the Authors (Required)):

The authors described a method that integrated targeted enrichment sequencing with single-cell (sc) RNA-seq. Although development of such methods is important, the authors did not show that their method provides sufficient advances over the existing methods (for example PMID: 31270458). For example, this study did not demonstrate a higher throughput of their method either in terms of the number of cells assayed or in terms of the number of targets profiled. Furthermore, the analyses presented in the study are shallow; problems are often glossed over; and evaluations of experimental parameters, like the volume of amplified cDNA as input, the number of PCR cycles and the sequencing depth are too simplistic. Finally, the text is sometimes hard to follow and the figures are not well-prepared.

Major issues:

1. The authors describe application of this method to enrich 5 genes CCL5, CD38, ICOSLG, S100A11 and TREX1. However, 2 of those (CD38 and ICOSLG) did not work. The authors provide only a superficial comment that they failed "likely due to inefficient primers." This is not acceptable - the authors need to provide guidelines as to which genes may or may not work in the assay and exactly why and what are the rules for the primer design for the method.
2. Related to above. Of the 3 remaining genes, the authors really use just one (TREX1) to characterize performance of the method in depth. One gene is highly insufficient to show that the method works.
3. The authors have shown that the method works only when targeting the 3' ends of cDNAs, this makes the choice of the targeted region less flexible.
4. The initial library prepared with 10x Chromium library preparation protocol is restricted to polyA+ RNA, or can also include polyA- RNA? If it is the former, the application will not be wide enough.
5. Lines 264-265. If optimization of the methods is required in each experiment, the application will be hindered. The authors should provide an optimized basic protocol or guideline for each step. For example, the protocol to generate cDNA, the amount and/or concentration of the input cDNA, number of PCR cycles, sequencing depth, primer design, and how these should be modified regarding the number of targets.

Additional comments:

6. In section "The standard and targeted method may require additional correction of barcodes and UMIs", please describe the targeting strategy, for example, how many primers were used to amplify the target, TREX1 transcripts.
7. Lines 143-147, using the volume as a metric here is a little strange: unless the Chromium library preparation protocol is standard and the original input is determined, the volume can not really guide the users. Other parameters should be used instead.
8. The first two paragraphs in the section "Technical parameters determine number and proportion of independent transcripts" resemble experimental notes. For example, "Six Chromium runs lead to five 40 ul samples containing amplified cDNA (one run failed)" ...
9. Lines 240-242, to support the opinion that "this clone may have decreased in size over time", more evidence needs to be provided, for example, to exclude the possibility that the difference detected was merely due to technical variation.
10. Lines 252-263. The authors described the advantages over another similar approach, they also should describe the shortcoming compared to that method.
11. Lines 268-269. The authors concluded that "more input cDNA increases the number of detected transcripts", however, it is quite possible that when the input cDNA exceeds certain amount, it would not further increase the number of detected transcripts but rather have the opposite effect. The authors should address this concern either by increasing the range of the tested input cDNA or by modifying the text.
12. Lines 315-316. This is a strong over estimation.

Comments to the text and figures:

13. Lines 56-58. Please try not to repeat the sentences in the abstract.
14. In Figure 2B-E, the colors are somehow confusing, need to be presented in a clearer way.
15. Figure 2, X axis, change "Number reads" to "Number of reads".

Reviewer #2 (Comments to the Authors (Required)):

In this study, the authors aim to evaluate and demonstrate a targeted approach for scRNA-seq, as well as the possibility of detecting previously known genotypes/variants using the same data. While the concept of overlaying single cell gene expression

with genotypes is not entirely new, it remains interesting. The authors present several metrics that may be of potential interest to readers who are considering similar experiments. However, since only a small number of target genes were analyzed, the scope of their study is limited, and they cannot provide any general recommendations. They provide an example of overlaying a cell-type-specific variant in cell clusters obtained from gene expression data, which is interesting. However, they were not able to provide any examples where variants have an effect on the gene expression. It would be helpful to see more examples demonstrating the usefulness of this approach.

I have several questions and concerns about this study:

Firstly, a recently published method called Constellation-Seq describes a similar approach for enriching targeted sequences. Therefore, the authors should cite this study and compare differences and similarities with their approach.

Secondly, I believe that the authors tend to oversell their method, especially in the abstract. For instance:

- In line 13 of the abstract, the authors claim to have developed a straightforward, targeted sequencing approach. However, I think that a two-step PCR is too simplistic to justify as a new method, particularly if it is very similar to Constellation-Seq. Instead, I suggest rephrasing it as "evaluated a straightforward, targeted sequencing approach."

- In line 15 of the abstract, the authors describe their method as "powerful." This statement should be rephrased.

- In line 16 of the abstract, the authors state that they describe a recommended data processing pipeline, but I could not find any pipeline or code description in the manuscript. Nor did I find any mention of a processing pipeline in the results section.

Thirdly, in line 20-21 of the abstract, the authors claim that they demonstrate how targeted sequencing can be used to simultaneously investigate the presence of somatic variants and their impact on affected cells. However, I could not find any description of the impact of variants on cells in the manuscript. Instead, the authors state that they did not observe any effects.

Finally, the last statement in the abstract suggests that the targeted sequencing extension can be used to efficiently detect cell type or state-dependent gene expression changes associated with genetic variation and allele-specific expression and alternative splicing, facilitating our understanding of the functional effects of genetic variation. However, the authors did not demonstrate this in the manuscript, so this statement should be rephrased accordingly.

Line 65 and forward: Is the approach biased to detect only variants in the 3'-end (or 5') ? Please clarify.

Line 101:'The pattern of the targeted method is more abrupt'.. I am not sure I understand what the authors mean here?

Line 98-102, Are you reporting the number of unique reads per UMI here? It is a bit unclear to me.

Line 101:'The pattern of the targeted method is more abrupt'.. Perhaps rephrase?

Figure 2. What is the main message the authors want to convey in this figure?. I suggest keeping those plots and move some to the supplementary. Moreover, axes labels are too small to read and I strongly recommend to use a another color gradient than grey

Figure 3. Nice demonstration of overlaying the genotype in the clusters. Please label the clusters with cell-type information in the UMAP.

It is not surprising that the amount of cDNA, number of PCR cycles, sequencing depth, and pooling of data from technical replicates will increase the number of detected targets. It would have been helpful to see more specific recommendations, such as the optimal number of PCR cycles. However, given the low number of targets analyzed in this study, it may not be possible to provide such general recommendations.

Reviewer #3 (Comments to the Authors (Required)):

The manuscript by Horebeek et al. describes a new capture method to genotype expressed single-cell transcripts. I have the following comments that could improve the manuscript.

- Multiple novel single-cell long-read whole transcriptome and targeted methods have been recently published. Please consider indicating that the novel targeted method described in this manuscript is based on short-read sequencing. Therefore, this method's read length (150bp) has major limitations in capturing alternative splice variants or SNV that don't lay close to the cDNA's 3' or 5' end.

- For figure 2, what is the percentage of on-target reads (TRES1) for the capture sequencing?

- What is the number of UMIs (ITSs) captured per cell for the standard method and the capture method for TRES1, CCL5 and S100A11 (Figure 3)? Does this change with increased cDNA, PCR cycles and sequencing depth?

- What was the yield of cDNA used for the different samples? So what is the amount of cDNA ng/ul for samples 4, 5 and 6 (Figure 3)?

- In the standard method, are CD38 and ICOSLG expressed? If it is, please comment on why capturing CD38 and ICOSLG with the new method in the discussion was impossible.

- Why is the overlap between molecules captured in libraries starting from the same cDNA sample so low (Supplementary Figure 3)? Mainly since amplified 10x cDNA was used as input.
- What is the correlation of UMIs per cell captured with the standard method and the new capture method (figure 4)?
- For figure 4, what is the gene expression of TREX1 in the T1 and T2 samples for the standard and capture methods?
- For the T2 sample in figure 4, more cells were detected with only the alternate TREX1 transcript (red). Were multiple UMIs detected of this alternate transcript per cell? What was the read depth per cell?

We thank the Reviewers for their useful feedback and suggestions. A subset of comments was addressed by including an additional dataset. As this resulted in substantial adaptations all over the manuscript, and for the sake of clarity, we do not highlight all the changes we made.

Reviewer #1

The authors described a method that integrated targeted enrichment sequencing with single-cell (sc) RNA-seq. Although development of such methods is important, the authors did not show that their method provides sufficient advances over the existing methods (for example PMID: 31270458). For example, this study did not demonstrate a higher throughput of their method either in terms of the number of cells assayed or in terms of the number of targets profiled. Furthermore, the analyses presented in the study are shallow; problems are often glossed over; and evaluations of experimental parameters, like the volume of amplified cDNA as input, the number of PCR cycles and the sequencing depth are too simplistic. Finally, the text is sometimes hard to follow and the figures are not well-prepared.

We regret that we were not able to convince the Reviewer of the strengths of our manuscript, but are grateful for the feedback given. We adapted the manuscript to address as many comments as feasible. We briefly address the points of criticism here, but more extensive information can be found below.

- *Advances over other methods: We do not want to claim our method is better than the Nam et al. method, as differences between both methods result in advantages as well as disadvantages for each method. As an example, our extension does not use transcript-specific spike-ins during reverse transcription (in contrast to Nam et al.). Thus, by design, we do not maximize the number of transcripts that can potentially be detected. However, this brings the advantage that we know we are not introducing bias in the standard method, and we also create more flexibility by being able to apply the targeted method on stored cDNA after analyzing the output from the standard method.*
- *Higher throughput: We adapted the manuscript to include measures such as number of cells and number of transcripts per cells more explicitly. We successfully pooled 3 targets for the systematic analysis, but have no data regarding the maximal number of targets with our approach. With Constellation-seq (Vallejo et al., 2021), up to 20 targets can be pooled in one amplification reaction. This method shows enough similarities to conclude that this finding is relevant for our method, but differs in the usage of linear amplification and does not aim at genotyping transcribed variants.*
- *Shallow and simplistic analyses: We have updated some of the analyses in line with the Reviewer Comments, obtaining more insights from our data. At the same time, we delineate our aims better, focusing on facilitating the choice of the technical parameters in future experiments. For this, we have created Supplementary Notes with what we consider the most important information.*
- *Text and figures: We hope the Reviewer can appreciate our extensive efforts to clarify both texts and figures.*

Major issues:

1. The authors describe application of this method to enrich 5 genes CCL5, CD38, ICOSLG, S100A11 and TREX1. However, 2 of those (CD38 and ICOSLG) did not work. The authors provide only a superficial comment that they failed "likely due to inefficient primers." This is not acceptable - the authors need to

provide guidelines as to which genes may or may not work in the assay and exactly why and what are the rules for the primer design for the method.

The nature of our approach does not give us a lot of flexibility during primer design. As the goal is to sequence over a transcribed variant of interest, primers should be located within a 150 bp region. In addition, the aspecific forward primer is fixed by design and in case of multiplexing, all primer pairs should function at the same annealing temperature. As with all PCR reactions, it is not always clear which experimental parameter is responsible for a non-functioning primer pair.

Regarding CD38 and ICOSLG specifically: we already observed during optimization of the individual primers that we were not able to identify the transcripts' sequences in the output of the Sanger sequencing. As we were not sure that Sanger sequencing was sufficiently sensitive and no better primer candidates could be designed, we decided to include these primers in the final experiment just to see whether the next-generation sequencing matched our Sanger sequencing data. The observed agreement between Sanger and NGS for these primers now validates the use of the (cheaper and faster) Sanger sequencing method during optimization. We included this information in our manuscript.

It is not possible to formulate specific guidelines as to which genes may or may not work, as the choice of transcribed variants of interest determines where the reverse primer should be located. However, we do provide clearer guidelines for creation of test cDNA (Supplementary note 1) and primer design and testing (Supplementary note 2).

2. Related to above. Of the 3 remaining genes, the authors really use just one (TREX1) to characterize performance of the method in depth. One gene is highly insufficient to show that the method works.

We included an additional dataset where we repeat the analyses when targeting IL7R. While the technical parameters were more optimal and the dataset does not require the additional correction based on Hamming distance, our other insights obtained from the TREX1 data are replicated in this dataset. The IL7R dataset is obtained from a much more abundant transcript and uses 5' sequencing, indicating that our original findings were generalizable, at least for these differences.

3. The authors have shown that the method works only when targeting the 3' ends of cDNAs, this makes the choice of the targeted region less flexible.

The new IL7R data uses 5' chemistry, demonstrating that both transcript ends can be used successfully.

4. The initial library prepared with 10x Chromium library preparation protocol is restricted to polyA⁺ RNA, or can also include polyA⁻ RNA? If it is the former, the application will not be wide enough.

Our extension is based on amplified cDNA generated during 10X Chromium library preparation and thus keeps the inherent limitations. As such, it is restricted to polyA⁺ RNA. When adapting the extension to scRNAseq methods other than 10X Chromium, it may be possible to target polyA⁻ RNA.

5. Lines 264-265. If optimization of the methods is required in each experiment, the application will be hindered. The authors should provide an optimized basic protocol or guideline for each step. For example, the protocol to generate cDNA, the amount and/or concentration of the input cDNA, number of PCR cycles, sequencing depth, primer design, and how these should be modified regarding the number of targets.

We supplemented the information for primer optimization present in the M&M with a step-by-step protocol in Supplementary notes 1 and 2.

Additional comments:

6. In section "The standard and targeted method may require additional correction of barcodes and UMIs", please describe the targeting strategy, for example, how many primers were used to amplify the target, TREX1 transcripts.

We added limited information to the Results section so it is easier for the reader to follow. More extensive information on primers and experimental set-up can be found in the Materials and Methods section.

7. Lines 143-147, using the volume as a metric here is a little strange: unless the Chromium library preparation protocol is standard and the original input is determined, the volume cannot really guide the users. Other parameters should be used instead.

The original Chromium library preparation protocols focus on volume as metric, which is why we also used this approach. With a limited amount of amplified cDNA available (only 40µl is generated), using some for determining the concentration means that less is available for other applications.

In the more recent Chromium library preparation protocols, they now do advise to measure concentration after cDNA amplification, so that up to 50 ng (maximum 20 µl) can be used for the standard gene expression library construction. In contrast, for the additional steps offered by 10X Genomics to enrich for TCR or Ig (BCR), the protocol states that you should proceed with 2 µl, independent of the concentration.

In line with this, we decided to express our recommendations in terms of volumes too.

We have concentration measurements available only for the IL7R data.

	P1	P2
Amplified cDNA	3.70 ng/µl	2.94 ng/µl
Input used to create targeted library	10 µl ≈ 37 ng	10 µl ≈ 29.4 ng
Targeted library	90.4 ng/µl	88.4 ng/µl

8. The first two paragraphs in the section "Technical parameters determine number and proportion of independent transcripts" resemble experimental notes. For example, "Six Chromium runs lead to five 40 ul samples containing amplified cDNA (one run failed)" ...

We rewrote and restructured the paragraph.

9. Lines 240-242, to support the opinion that "this clone may have decreased in size over time", more evidence needs to be provided, for example, to exclude the possibility that the difference detected was merely due to technical variation.

We meant to offer the decrease in clone size as a possible explanation, as we can indeed not exclude other explanations such as technical variation. We reformulated it to "suggesting this clone may have decreased in size over time."

10. Lines 252-263. The authors described the advantages over another similar approach, they also should describe the shortcoming compared to that method.

As shortcomings of one method can be formulated as strengths of the other, we highlighted the strengths of each in the first paragraph of the discussion. In the one to last paragraph of the discussion, we discuss the remaining limitations of our approach.

11. Lines 268-269. The authors concluded that "more input cDNA increases the number of detected

transcripts", however, it is quite possible that when the input cDNA exceeds certain amount, it would not further increase the number of detected transcripts but rather have the opposite effect. The authors should address this concern either by increasing the range of the tested input cDNA or by modifying the text.

We believe there are two different actors at play that may have caused confusion: (1) the amount of amplified cDNA that is used as input, and (2) the number of cells in the sample.

- 1. Following the 10X Chromium protocol, 40 μ l amplified cDNA is created for each sample. This means that the maximum amount of amplified cDNA that can be used as input for targeted sequencing is systematically capped at 40 μ l. To reflect the actual experimental situation when combining the standard and targeted method, we included maximum 20 μ l as input in the systematic analysis. As part of our systematic analysis, we have used 35 μ l amplified cDNA to make three targeted libraries (starting from 5 μ l, 10 μ l or 20 μ l amplified cDNA). We observe that the overlap between detected transcripts from the 5 μ l, 10 μ l and 20 μ l input libraries is limited and thus believe that even a library made from all 40 μ l amplified cDNA will lead to more detected transcripts than one made from less. Following the standard 10x experimental set up, combined with the targeted sequencing method as we describe it, we are still convinced that more input cDNA increases the number of detected transcripts. Only if for some reason, very extensive PCR is done to create the amplified cDNA and the samples are sequenced deeply, saturation may occur and increasing input will not lead to additional transcripts being detected.*
- 2. Another input variable that may differ is number of cells in the sample. When increasing the number of cells, a decrease in transcripts per cell is expected when keeping all other parameters identical. This is outside the scope of our systematic analysis.*

12. Lines 315-316. This is a strong over estimation.

We replaced 'this method provides a big step forward' by 'this method offers a helpful step forward'.

Comments to the text and figures:

13. Lines 56-58. Please try not to repeat the sentences in the abstract.

We rewrote the abstract and changed the repeated sentence.

14. In Figure 2B-E, the colors are somehow confusing, need to be presented in a clearer way.

We completely remade Figure 2. We now focus on two example datasets and move the other samples to Supplementary Data.

15. Figure 2, X axis, change "Number reads" to "Number of reads".

This was originally chosen to be able to increase label size. With the remaking of the figure, we went back to the full version of the axis label.

Reviewer #2

In this study, the authors aim to evaluate and demonstrate a targeted approach for scRNA-seq, as well as the possibility of detecting previously known genotypes/variants using the same data. While the concept of overlaying single cell gene expression with genotypes is not entirely new, it remains interesting. The authors present several metrics that may be of potential interest to readers who are considering similar

experiments. However, since only a small number of target genes were analyzed, the scope of their study is limited, and they cannot provide any general recommendations. They provide an example of overlaying a cell-type-specific variant in cell clusters obtained from gene expression data, which is interesting. However, they were not able to provide any examples where variants have an effect on the gene expression. It would be helpful to see more examples demonstrating the usefulness of this approach.

We thank the Reviewer for this accurate summary of our manuscript. We limited our proof-of-concept investigation of the effect of somatic variation in multiple sclerosis patients to one variant, which did not affect gene expression. We added another example (IL7R data) where we look at and detect allele-specific expression.

I have several questions and concerns about this study:

Firstly, a recently published method called Constellation-Seq describes a similar approach for enriching targeted sequences. Therefore, the authors should cite this study and compare differences and similarities with their approach.

We thank the Reviewer for pointing out this study. It seems this publication slipped through our literature search.

Constellation-Seq is a very elegant method, where linear amplification is used to address some of the struggles we are facing with unequal amplification. As Constellation-Seq is not demonstrated for genotyping of transcripts, we did not add it to the comparison with the Nam et al. method. Instead, we refer to it when discussing the effect of number of PCR cycles as linear amplification seems like an efficient alternative approach to avoid unequal amplification. We highlight the characteristics that are most relevant to our manuscript (linear amplification and 20 targets simultaneously).

Secondly, I believe that the authors tend to oversell their method, especially in the abstract. For instance:

- In line 13 of the abstract, the authors claim to have developed a straightforward, targeted sequencing approach. However, I think that a two-step PCR is too simplistic to justify as a new method, particularly if it is very similar to Constellation-Seq. Instead, I suggest rephrasing it as "evaluated a straightforward, targeted sequencing approach."
- In line 15 of the abstract, the authors describe their method as "powerful." This statement should be rephrased.
- In line 16 of the abstract, the authors state that they describe a recommended data processing pipeline, but I could not find any pipeline or code description in the manuscript. Nor did I find any mention of a processing pipeline in the results section.

We rewrote the abstract to accommodate these comments.

Thirdly, in line 20-21 of the abstract, the authors claim that they demonstrate how targeted sequencing can be used to simultaneously investigate the presence of somatic variants and their impact on affected cells. However, I could not find any description of the impact of variants on cells in the manuscript. Instead, the authors state that they did not observe any effects.

We agree that this was an unfortunate formulation and rephrased the sentence.

Finally, the last statement in the abstract suggests that the targeted sequencing extension can be used to efficiently detect cell type or state-dependent gene expression changes associated with genetic variation and allele-specific expression and alternative splicing, facilitating our understanding of the functional

effects of genetic variation. However, the authors did not demonstrate this in the manuscript, so this statement should be rephrased accordingly.

We rephrased the abstract, referring only to our demonstrated proof-of-principles: "Finally, we demonstrate how targeted sequencing can be used in two contexts: (1) simultaneous investigation of the presence of a somatic variant and its potential impact on the transcriptome of affected cells, and (2) evaluation of allele-specific expression of a germline variant in ad hoc cell subsets. Through these and a myriad of comparable applications, our targeted sequencing extension has the potential to improve our understanding of functional effects caused by genetic variation."

Line 65 and forward: Is the approach biased to detect only variants in the 3'-end (or 5') ? Please clarify.
We have successfully implemented the targeted sequencing approach for variants up to 1,000 bp from the transcript end. Longer fragments are expected to work as well, but may require some adaptations such as longer extension time during PCR. Additionally, possibilities for multiplexing during sequencing may be limited, as shorter fragments bind easier to Illumina sequencers and may thus outcompete the longer ones. Nam et al. has a circular method to remove the superfluous middle part of the construct, providing an alternative method to include variants further from the transcript ends.

Line 101:'The pattern of the targeted method is more abrupt'.. I am not sure I understand what the authors mean here?

We meant that when looking at the plots of number of reads per tagging sequence, we observe an exponential decrease overall, but that some datasets have substantially more tagging sequences covered by one read than expected based on an exponential decrease.

With the addition of the extra dataset, we rewrote the paragraph. Figure 2 now shows an example of a logarithmic decrease and an example with substantially more tagging sequences covered by one read. The plots from all datasets can be found in Supplementary figures 2-9.

Line 98-102, Are you reporting the number of unique reads per UMI here? It is a bit unclear to me.

We reported the percentage of tagging sequences (defined as combination of cellular barcode and UMI), that was supported by only one unique read. With the addition of the extra dataset, we removed the percentages in the text, but the distributions can be observed in the plots (Figure 2 + Supplementary figures 2-9).

Line 101:'The pattern of the targeted method is more abrupt'.. Perhaps rephrase?

With the addition of the new dataset, we rewrote the whole paragraph, keeping in mind this comment.

Figure 2. What is the main message the authors want to convey in this figure? I suggest keeping those plots and move some to the supplementary. Moreover, axes labels are too small to read and I strongly recommend to use another color gradient than grey

We simplified Figure 2 by focusing on two samples (targeted sequencing from TREX1 at T1 and targeted sequencing from IL7R P2) and reporting all data in Supplementary figures 2-5. By focusing on two samples, we can show the two trends we notice across datasets (exponential decrease and occasionally an abundance of tagging sequences with 1 supporting read). We also updated the lay-out of the figure to improve readability.

Figure 4. Nice demonstration of overlaying the genotype in the clusters. Please label the clusters with cell-type information in the UMAP.

We have updated Figure 4 and included cell type annotations in the UMAP plot.

It is not surprising that the amount of cDNA, number of PCR cycles, sequencing depth, and pooling of data from technical replicates will increase the number of detected targets. It would have been helpful to see more specific recommendations, such as the optimal number of PCR cycles. However, given the low number of targets analyzed in this study, it may not be possible to provide such general recommendations. *With our current data, we can only recommend to use as much input cDNA as possible, to sequence deep enough and that 20 PCR cycles gives better results than 10 or 15 PCR cycles (Supplementary note 2). We also observe that additional sequencing of the same library or the creation of new libraries from the same cDNA may increase the amount of information that can be obtained. We formulated these recommendations in Supplementary note 2.*

Reviewer #3

The manuscript by Van Horebeek et al. describes a new capture method to genotype expressed single-cell transcripts. I have the following comments that could improve the manuscript.

We thank the Reviewer for the suggestions, which are addressed below.

- Multiple novel single-cell long-read whole transcriptome and targeted methods have been recently published. Please consider indicating that the novel targeted method described in this manuscript is based on short-read sequencing. Therefore, this method's read length (150bp) has major limitations in capturing alternative splice variants or SNV that don't lay close to the cDNA's 3' or 5' end.

We have added that the targeted sequencing method uses short-read sequencing in the description of the method in the Results section.

- For figure 2, what is the percentage of on-target reads (TREX1) for the capture sequencing?

We have added the percentage of on-target reads to Table 1. With the captured sequencing, >80% of reads are mapped to the expected region.

- What is the number of UMIs (iTSs) captured per cell for the standard method and the capture method for TREX1, CCL5 and S100A11 (Figure 3)? Does this change with increased cDNA, PCR cycles and sequencing depth?

We have added this information for the different datasets. For TREX1 and IL7R, this data is now included in Table 1. For the systematic analysis, we added the plots to Figure 3 panels A-C.

- What was the yield of cDNA used for the different samples? So what is the amount of cDNA ng/ul for samples 4, 5 and 6 (Figure 3)?

As the original 10X Chromium library preparation protocols focused on volume and only a limited volume of amplified cDNA is available, we did not measure the cDNA concentrations for most datasets. Only for the IL7R dataset do we have the information available.

	P1	P2
Amplified cDNA	3.70 ng/ul	2.94 ng/ul

Input used to create targeted library	10 μ l \approx 37 ng	10 μ l \approx 29.4 ng
Targeted library	90.4 ng/ μ l	88.4 ng/ μ l

- In the standard method, are CD38 and ICOSLG expressed? If it is, please comment on why capturing CD38 and ICOSLG with the new method in the discussion was impossible.

We did not run the standard method on the samples from the systematic analysis, as not enough amplified cDNA was available to do both. We do have data from the standard method from B-cells from other untreated MS patients. Success and failure of primers did not correspond to expression levels in these samples or in B cells from the Human Protein Atlas (proteinatlas.org), indicating that low expression levels is an unlikely explanation for failed capturing of CD38 and ICOSLG.

	MS-A (UMIs/10 ⁶ UMIs)	MS-B (UMIs/10 ⁶ UMIs)	Protein Atlas – B cells (nTPM)
CCL5	6.49	7.83	43.1
CD38	7.85	16.84	21.4
ICOSLG	18.41	28.73	3.7
S100A11	33.78	47.81	186.6
TREX1	28.72	29.58	1.8

[Start overlap with reply to Reviewer 1 Comment 1]

The nature of our approach does not give us a lot of flexibility during primer design. As the goal is to sequence over a transcribed variant of interest, primers should be located within a 150 bp region. In addition, the aspecific forward primer is fixed by design and in case of multiplexing, all primer pairs should function at the same annealing temperature. As with all PCR reactions, it is not always clear which experimental parameter is responsible for a non-functioning primer pair.

Regarding CD38 and ICOSLG specifically: we already observed during optimization of the individual primers that we were not able to identify the transcripts' sequences in the output of the Sanger sequencing. As we were not sure that Sanger sequencing was sufficiently sensitive and no better primer candidates could be designed, we decided to include these primers in the final experiment just to see whether the next-generation sequencing matched our Sanger sequencing data. The observed agreement between Sanger and NGS for these primers now validates the use of the (cheaper and faster) Sanger sequencing method during optimization. We included this information in our manuscript.

We cannot formulate specific guidelines as to which genes may or may not work, as the choice of transcribed variants of interest determines where the reverse primer should be located. However, we do provide clearer guidelines for creation of test cDNA (Supplementary note 1) and primer design and testing (Supplementary note 2).

- Why is the overlap between molecules captured in libraries starting from the same cDNA sample so low (Supplementary Figure 3)? Mainly since amplified 10x cDNA was used as input.

We identified two factors that may contribute to the limited overlap between molecules captured in libraries created from the same cDNA. First, a low amount of amplification is recommended by 10X, to not introduce too much bias and to avoid errors. Therefore, some transcripts may not be very abundant and thus restricted to one library. Second, the sequencing depth also influences the overlap, as can be observed in Figure 3E. When sequencing depth increases, the overlap between molecules from different replicates increases too.

- What is the correlation of UMIs per cell captured with the standard method and the new capture method (figure 4)?

We observe a positive correlation between UMIs/cell in standard and targeted method, in line with expectations. We show the data and include the correlation information in Supplementary figures 12 and 13.

- For figure 4, what is the gene expression of TREX1 in the T1 and T2 samples for the standard and capture methods?

We included the (normalized) gene expression levels as measured with the standard methods in Supplementary figure 12, panel C. Panel B in the same figure shows the number of UMIs/cell for both the standard and targeted method.

- For the T2 sample in figure 4, more cells were detected with only the alternate TREX1 transcript (red). Were multiple UMIs detected of this alternate transcript per cell? What was the read depth per cell?

The sparsity due to the intermediate TREX1 expression levels in CD8⁺ T lymphocytes is responsible for detecting more cells with alternate genotype call than with heterozygous genotype call. For the targeted data from T2, we observe the following distribution in number of transcripts per cell:

Legend:

Full red line: ref transcripts/cell in cells with ref call

Full blue line: alt transcripts/cell in cells with alt call

Dashed red line: ref transcripts/cell in cells with het call

Dashed blue line: alt transcripts/cell in cells with het call

For these transcripts (T2, targeted), a skewed distribution (similar to Supplementary figure 3) is observed, with most transcripts being supported by 1 read. Reference transcripts are supported by 1 [1, 1433] (median [min, max]) reads and alternative transcripts are supported by 1 [1, 1439] (median [min, max]) reads.

Part of this information can also be found in Table 1 and Supplementary Figures 12 and 13.

August 7, 2023

RE: Life Science Alliance Manuscript #LSA-2023-01971-TR

Prof. Benedicte Dubois
KU Leuven
Herestraat 49 box 1022
Leuven 3000
Belgium

Dear Dr. Dubois,

Thank you for submitting your revised manuscript entitled "A targeted sequencing extension for transcript genotyping in single-cell transcriptomics". We would be happy to publish your paper in Life Science Alliance pending final revisions necessary to meet our formatting guidelines.

- please upload your main manuscript text as an editable doc file
- please upload your main and supplementary figures as single files
- please add ORCID ID for the corresponding author--you should have received instructions on how to do so
- please add the Twitter handle of your host institute/organization as well as your own or/and one of the authors in our system
- please remove the graphical abstract from the manuscript text and upload it separately with the file designation "Graphical abstract"
- please add an Author Contributions section to your main manuscript text
- please upload your Tables in editable .doc or excel format
- please remove your figures from the manuscript text; all figure legends should only appear in the main manuscript file
- please add your main, supplementary figure, and table legends to the main manuscript text after the references section
- please include callouts for all panels in the supplementary figures in your manuscript text
- please add callouts for Figures 4A-C and 5A to your main manuscript text
- please rename your "Data Access" section to "Data Availability" GSE223704 dataset should be made publicly accessible at this point, and please update the Data Availability statement to remove the Reviewer access information
- the Supplementary Notes should be combined and uploaded as a single Supplemental Material file. You can remove the "Supplementary Note" designation, and just keep the heading names of each individual section. Please update the callouts in the paper to "Supplemental Material". Please also indicate in the Data Availability statement that code is available in the Supplemental Material, and explain there what it is for.

A. FINAL FILES:

-- Summary blurb (enter in submission system): A short text summarizing in a single sentence the study (max. 200 characters including spaces). This text is used in conjunction with the titles of papers, hence should be informative and complementary to

the title. It should describe the context and significance of the findings for a general readership; it should be written in the present tense and refer to the work in the third person. Author names should not be mentioned.

B. MANUSCRIPT ORGANIZATION AND FORMATTING:

Sincerely,

Reviewer #1 (Comments to the Authors (Required)):

The authors did not clearly indicate the changes they made in the manuscript that address each specific comment. They answered but did not really address several of the reviewer's concerns (comments 1, 4, 5 for example). Although they tried to improve the manuscript by adding one additional dataset that can repeat their previous result, the overall advance, reliability, and applicability of this method are not improved and thus remain not convincing. As long as the authors can increase the throughput and sensitivity of the method and can show that the protocol they provided can be reliably used for multiple targets, the manuscript could be further considered.

Reviewer #2 (Comments to the Authors (Required)):

The authors have addressed most of my concerns and those of the other reviewers, and they have made significant improvements to the manuscript. In my opinion this now warrants publication.

Reviewer #3 (Comments to the Authors (Required)):

My concerns have been adequately addressed.

August 29, 2023

RE: Life Science Alliance Manuscript #LSA-2023-01971-TRR

Prof. Benedicte Dubois
KU Leuven
Herestraat 49 box 1022
Leuven 3000
Belgium

Dear Dr. Dubois,

Thank you for submitting your Methods entitled "A targeted sequencing extension for transcript genotyping in single-cell transcriptomics". It is a pleasure to let you know that your manuscript is now accepted for publication in Life Science Alliance. Congratulations on this interesting work.

DISTRIBUTION OF MATERIALS:

Again, congratulations on a very nice paper. I hope you found the review process to be constructive and are pleased with how the manuscript was handled editorially. We look forward to future exciting submissions from your lab.

Sincerely,
